# Potential Role of Anti-Müllerian Hormone in Regulating Seasonal Reproduction in Animals: The Example of Males

**DOI:** 10.3390/ijms24065874

**Published:** 2023-03-20

**Authors:** Kang An, Baohui Yao, Yuchen Tan, Yukun Kang, Junhu Su

**Affiliations:** 1Key Laboratory of Grassland Ecosystem, Ministry of Education, College of Grassland Science, Gansu Agricultural University, Lanzhou 730070, China; 2Massey University Research Centre for Grassland Biodiversity, Gansu Agricultural University, Lanzhou 730070, China

**Keywords:** anti-Müllerian hormone, seasonal reproduction, testosterone, reproduction suppression, testes

## Abstract

Seasonal reproduction is a survival strategy by which animals adapt to environmental changes to improve their fitness. Males are often characterized by a significantly reduced testicular volume, indicating that they are in an immature state. Although many hormones, including gonadotropins, have played a role in testicular development and spermatogenesis, research on other hormones is insufficient. The anti-Müllerian hormone (AMH), which is a hormone responsible for inducing the regression of Müllerian ducts involved in male sex differentiation, was discovered in 1953. Disorders in AMH secretion are the main biomarkers of gonadal dysplasia, indicating that it may play a crucial role in reproduction regulation. A recent study has found that the AMH protein is expressed at a high level during the non-breeding period of seasonal reproduction in animals, implying that it may play a role in restricting breeding activities. In this review, we summarize the research progress on the *AMH* gene expression, regulatory factors of the gene’s expression, and its role in reproductive regulation. Using males as an example, we combined testicular regression and the regulatory pathway of seasonal reproduction and attempted to identify the potential relationship between AMH and seasonal reproduction, to broaden the physiological function of AMH in reproductive suppression, and to provide new ideas for understanding the regulatory pathway of seasonal reproduction.

## 1. Introduction

Reproduction is a biological process with high energy costs. To convert limited resources into maximum adaptation, many organisms have evolved adaptive strategies, such as seasonal reproduction. This refers to a breeding strategy developed by animals living in non-tropical regions in the face of dramatic changes in environmental factors throughout the year [1]. Compared with perennial estrous, seasonal reproduction restricts breeding activities to a specific time of the year that is most conducive to their survival and that of their offspring, ensuring a high success rate of breeding [2]. These animals can accurately predict seasonal changes and adjust their physiological status by sensing periodic changes in environmental signals, such as light, temperature, precipitation, and food resources [3]. Melatonin (MLT), which is secreted by the pineal gland of the brain, has the characteristics of high secretion at night and light inhibition during the day. With the periodic changes in the lengths of day and night throughout the year, MLT also has seasonal differences in secretion, which is considered to be the main mechanism for animals to obtain seasonal periodic change signals [4]. The secreted MLT binds to a receptor on the hypothalamic–pituitary–gonadal (HPG) axis, which provides a seasonal change signal to the animal body, thus regulating periodic variation in breeding activity [5]. In studies on seasonal reproduction, the role of the gonadotropin-releasing hormone (GnRH), gonadotropins, and MLT in reproduction regulation has been confirmed [1,6], but research on other hormones is insufficient. The anti-Müllerian hormone (AMH), also known as the Müllerian-inhibiting substance, is a dimeric glycoprotein that is part of the transforming growth factor beta (TGF-β) superfamily, together with inhibin, activin, bone morphogenetic proteins, and growth differentiation factors [7,8]. The TGF-β family members play a role in several biological processes, including follicle development, spermatogenesis, cell proliferation, and apoptosis [9]. A recent study found that the AMH protein is highly expressed during the non-breeding season [10], suggesting that it may play a role in restricting seasonal breeding activities; however, the relationship between AMH and seasonal reproduction in animals is unknown.

In this review, we attempt to identify the relationship between AMH and seasonal reproduction, with the goal of broadening knowledge on the physiological function of AMH in reproductive suppression while also providing new ideas for understanding the regulatory pathway of seasonal reproduction.

## 2. *AMH* Gene Expression

During the early stages of mammalian embryonic development, males and females have two pairs of ducts: Wolffian ducts and Müllerian ducts [11]. Wolffian ducts differentiate and develop into the male epididymis, vas deferens, and seminal vesicles, and Müllerian ducts differentiate and develop into the female fallopian tube, uterus, and upper vagina [12]. In the absence of AMH, the Müllerian ducts of male embryos develop into female reproductive organs, resulting in the existence of female reproductive organs in male individuals and failure of sex differentiation [13].

Immature Sertoli cells (SCs) in the testes and granulosa cells in the ovaries produce AMH. During embryonic development in humans, the *AMH* gene starts to be expressed by immature SCs in XY embryos when the testes differentiate from the gonadal ridge, and it maintains a high level of secretion until the beginning of puberty [13]. In mice (*Mus musculus*), the *AMH* gene begins to be expressed in embryos 12.5 d after copulation [14]. The *AMH* gene expression in the early embryonic stage does not depend on the stimulation of external factors, as shown by a study that found that the sex-determining region Y (*SRY*) gene was expressed in the immature SCs that first appeared [15], and the SRY box 9 (*SOX9*) was up-regulated by the *SRY* gene, triggering the expression of the *AMH* gene by combining with specific elements in the proximal promoter of the *AMH* gene [16]. Subsequently, steroidogenic factor 1 (*SF1*) [17], GATA binding protein 4 (*GATA4*) [18], and Wilms’ tumor 1 (*WT1*) [19] transcription factors play a synergistic role in further increasing the expression level of the *AMH* gene. With the activation of the HPG axis later in the pregnancy, the luteinizing hormone (LH) and follicle-stimulating hormone (FSH) secreted by the pituitary gland play a crucial role in the regulation of *AMH* gene expression.

### 2.1. FSH Stimulates the Expression of AMH Gene

During later stages of embryonic development and after birth, AMH is produced by testicular SCs and can be further induced by FSH [20]. FSH stimulates SCs to secrete AMH through two different mechanisms: (1) FSH induces SCs to proliferate, increasing the number of cells to increase the level of AMH secretion; (2) FSH up-regulates the expression level of the *AMH* gene in SCs [20,21]. A study found that FSH in SCs sends signals through the cyclic adenosine monophosphate (cAMP) and protein kinase A (PKA) pathways to induce an increase in *AMH* gene expression. Although the *AMH* gene promoter lacks a classical cAMP response element, some transcription factor sites exist in the *AMH* gene promoter that respond to cAMP and regulate the expression of the *AMH* gene [22]. The specific regulatory mechanism is shown in Figure 1, where FSH binding to the FSH receptor (FSHR) induces an increase in cAMP levels, activating several protein kinases. The main kinase responsible for up-regulating *AMH* gene transcription is PKA, which induces the translocation of *SOX9*, *SF1*, *GATA4*, adaptor protein 2 (*AP2*), and nuclear factor kappa-B (*NF-κB*) transcription factors into the nucleus and binds to the promoter region to jointly promote the expression of the *AMH* gene. In addition, cAMP promotes the proliferation of SCs and increases the number of cells required to produce AMH. Therefore, AMH is regarded as a biomarker of FSH in the testes before puberty.

### 2.2. Estrogen Stimulates the Expression of AMH Gene

Estrogen is a steroid hormone synthesized by androgens and catalyzed by aromatase, which is a member of the cytochrome P450 superfamily encoded by only one gene, *CYP19A1*, in humans [23]. Androgen insensitivity syndrome (AIS) in humans is caused by a defect in androgen synthesis or a mutation in the androgen receptor (AR) gene and causes the level of AMH to not decrease during puberty [24]. Although serum FSH levels are decreased in Peutz–Jeghers syndrome (PJS), the AMH level remains high [25]. Interestingly, there are high estrogen levels in the serum of patients with AIS and PJS, implying that estrogen may promote the expression of the *AMH* gene. Estradiol (E2) is the most important and most biologically active estrogen, and studies have confirmed that it regulates the expression of the *AMH* in the ovary [26,27]. In the human AMH promoter, a consensus sequence of estrogen response elements (ERE) is present at −1782 bp [28]. Valeri et al. used experimental mouse models to confirm that E2 up-regulates the expression of the *AMH* gene in pre-pubertal and pubertal testes [27]. They believed that E2, mainly through estrogen receptor α (ERα) and the G-protein-coupled estrogen receptor (GPER), up-regulates the expression of the *AMH* gene and promotes the increase in the testicular AMH secretion level [27]. Another potential mechanism for increasing the AMH secretion level in the testes is E2 binding to ERα on the cell membrane to induce cell proliferation through the PI3K/Akt and GPER/MAPK signaling pathways, thereby increasing the number of cells producing AMH [27].

### 2.3. Androgen Inhibits the Expression of AMH Gene

Testosterone is the main androgen and regulates spermatogenesis in the testes. The synthesis of testosterone in testicular Leydig cells (LCs) is mainly divided into three steps: (1) LH binds to the LH receptor (LHR) on testicular LCs to induce the expression of the steroidogenic acute regulatory (StAR) protein and translocate cholesterol into the mitochondria; (2) cholesterol in the mitochondria is catalyzed by the CYP11A1 enzyme to generate pregnenolone (PREG); (3) PREG is transported to the smooth endoplasmic reticulum, regulated by two intermediates, progesterone and 17-OH-progesterone, and it is enzymatically hydrolyzed by 3β-HSD to androstenedione, which is then subjected to 17β-HSD to produce testosterone [29]. Testosterone secreted by LCs enters the seminiferous tubules through diffusion and binds to the AR to play a physiological role. Although the level of human serum AMH gradually decreases during the beginning of puberty and remains low throughout, the corresponding level of serum testosterone continuously increases [30], suggesting that testosterone has a negative regulatory effect on the expression of the *AMH* gene. In a study on precocious children, it was found that, although the FSH secretion level was normal, the increase in testosterone levels was always accompanied by a decrease in AMH levels [31]. It is interesting that the levels of AMH and testosterone increase synchronously in newborns, but it has been discovered that AR only exists in the LCs and peritubular cells of newborns and is not expressed in SCs [32]. Furthermore, in a study of SC-specific knockout mice, it was found that the presence of AR was required for the reduction in AMH levels [33]. These results show that the negative regulation of AMH by testosterone is mediated by AR and that androgen inhibition of *AMH* gene expression exceeds the stimulation of FSH.

Although no AR binding site exists in the promoter of the *AMH* gene [34], previous studies have found that the *NF-κB* transcription factor is negatively regulated by AR [35]. The *NF-κB* is considered to be a widely expressed transcription factor that can be activated in a variety of cell types, especially immune cells [36], and has been proven to up-regulate the expression of various genes involved in the mammalian immune and inflammatory response in immune cells. However, the combination of glucocorticoid and the glucocorticoid receptor (GR) is an effective inhibitor of immune response and inflammation, implying that GR and *NF-κB* transcription factors have an antagonistic effect [36]. In terms of structure and sequence specificity, AR is closely related to GR. Transient co-transfection experiments have also confirmed that AR and *NF-κB* transcription factors have an antagonistic effect [35], which may explain the inhibition of AR on *AMH* gene expression.

Recently, in another study on the inhibition of *AMH* gene expression by AR, androgens were found to have a direct negative regulatory effect on the transcriptional activity of the *AMH* gene promoter in mouse pre-pubertal SCs [37]. In the presence of testosterone and AR, the activity of the *AMH* gene promoter was strongly inhibited. Through site-directed mutagenesis and chromatin immunoprecipitation analysis, it was found that androgen-mediated inhibition was located on the binding site of *SF1* in the proximal promoter of the *AMH* gene. The authors hypothesized that AR and *SF1* have two possible mechanisms of action that inhibit the expression of the *AMH* gene (Figure 2): (1) Blockage by competition, which might be the binding site of AR and *SF1* competing for the *AMH* gene promoter to prevent *SF1*-mediated gene up-regulation and allow androgens to inhibit the AMH promoter activity. (2) Blockage by interaction, which may involve the AR binding to a site on *SF1*. This mutual interference can cause *SF1* to change conformation or interact with other activating elements, preventing *SF1* from up-regulating the promoter of the *AMH* gene [37].

## 3. Research of AMH in Reproduction Regulation

As early as the 1940s, researchers found that a testicular factor other than testosterone induced the regression of the Müllerian ducts during the development of male embryos and was later identified as AMH, which plays a critical role in the sex differentiation of fetuses [11,38]. Since then, numerous physiological functions of AMH have continuously been determined. According to research, AMH secretion disorder is the main biomarker of gonadal dysplasia, such as delayed puberty, premature puberty, and AIS [39]. These symptoms indicate that AMH may affect the growth and development of gonadal tissues. In females, the AMH level reflects the ovarian follicle reserve and serves as a biomarker for infertility treatment, ovarian aging, granulosa cell tumors, and polycystic ovary syndrome [40,41]. A previous study has shown that AMH inhibits the development of mouse ovaries during the embryonic period [42]. Recently, Meisohn et al. treated newborn mice with AMH and applied single-cell sequencing (scRNA-seq) methods to the ovary on the sixth day, thereby revealing the specific mechanism by which AMH inhibits the early growth and development of ovaries [43]. In males, early studies have shown that AMH affects spermatogenesis and negatively regulates the differentiation and function of testicular LCs [44,45]. Recently, Rehman et al. studied the effect of AMH on the proliferation of mouse testes’ SCs at the cellular level and found that a high concentration of AMH induced the expression of apoptosis genes and an increase in the apoptosis rate [46], which showed that AMH can affect reproduction at the testicular tissue level. In addition, AMH may play a regulatory role in the HPG axis during reproduction. Polycystic ovarian syndrome (PCOS) is a common symptom of female infertility [47]. The main sign is that the ratio of LH to FSH in the serum is higher than normal; at the same time, the level of circulating AMH hormone is higher in patients with PCOS [48], indicating that the AMH secretion disorder is a potential cause of the imbalance in the LH and FSH ratio. Cimino et al. found that AMH could stimulate the activity of GnRH neurons in the hypothalamus, resulting in an increase in the level of LH secretion in the pituitary gland, providing a scientific reference for the treatment of PCOS [49]. These existing studies have shown a broader physiological function of AMH in the field of reproduction, providing a scientific basis for studying the AMH regulatory role in seasonal reproduction (Figure 3).

## 4. Seasonal Reproduction Regulation Pathway

During seasonal changes, the photoperiod is always highly stable and is considered the main environmental factor regulating seasonal reproduction [50]. The level of MLT secreted by the pineal gland of the brain is strictly controlled by light. By binding to receptors on the HPG axis, MLT regulates seasonal reproduction and is involved in the physiological activities of various organs, tissues, and cells in the animal body. There are two types of high-affinity MLT receptors in mammals: melatonin receptor 1a (MTNR1a) and melatonin receptor 1b (MTNR1b). A previous study found that MTNR1a is highly expressed in cells that produce thyroid-stimulating hormone (TSH) in the pars tuberalis [51], indicating that MLT can affect the synthesis and secretion of TSH. In ependymal cells of the medial basal hypothalamus (MBH), TSH can regulate the conversion of type 2 deiodinase (Dio2) and type 3 deiodinase (Dio3), where Dio2 is an activator of thyroid hormone (TH) and Dio3 is a deactivator [52]. TH exists in two main forms in mammals such as humans, rats, and mice: triiodothyronine (T3) and tetraiodothyronine (T4) [53]. Dio2 catalyzes the conversion of inactive T4 to active T3, and Dio3 catalyzes the conversion of T4 to T3; thus, Dio2 and Dio3 jointly regulate the ratio of T3 and T4 to regulate the hormone levels of TH in the hypothalamic MBH [54]. Secreted TH binds to the corresponding receptors on the median eminence (ME) of the MBH of the hypothalamus for biological function, which is the projection of GnRH neuron endings on the ME. A high concentration of TH can induce morphological changes in GnRH nerve endings and the glial cell processes around them, and contact between GnRH nerve endings and the capillary basement membrane facilitates the release of GnRH into the blood. In addition to the TSH/TH pathway, which induces the hypothalamus to release GnRH, MLT can target kisspeptin and RF amide-related peptides to activate the HPG axis and regulate the activity of GnRH neurons to affect the release of GnRH [55].

The periodic release of GnRH in the hypothalamus stimulates the synthesis and secretion of LH and FSH in anterior pituitary cells [56]. By binding to the LHR and FSHR in the gonad tissue, LH and FSH perform their biological functions and are important communication factors among the hypothalamus, pituitary, and gonads. In males, LHR and FSHR are located in the LCs and SCs of the testicular tissue, respectively [57]. To regulate the occurrence of steroids and promote the synthesis of testosterone and E2, LHR on LCs binds to LH secreted by the pituitary gland; FSH regulates the synthesis and secretion of the androgen-binding protein, inhibin, AMH, and other SC secretions by binding to receptors, thereby jointly regulating testicular tissue development and spermatogenesis.

## 5. Testicular Regression of Seasonal Reproduction

Before puberty, SCs are the largest and most active group of cells in the human testes, and FSH stimulates the proliferation of immature SCs, causing a moderate increase in testicular volume [58]. After puberty, the HPG axis is activated, and testicular LCs and SCs jointly regulate the differentiation of germ cells (GCs) to produce mature sperm cells, thereby promoting a large increase in testicular volume [39]. In seasonal reproduction, animals spend the majority of the year in a static state of breeding and discontinuation of spermatogenesis, accompanied by the low levels of testosterone produced by the testicles, which is the cause of testicular regression or atrophy during the non-breeding season [59]. In animals such as the bank vole (*Clethyonomys glaeolus*) [60], prairie dog (*Cynomoys ludovicianus*) [61], and Syrian hamster (*Mesocricetus auratus*) [62], spermatogenesis is prevented during the non-breeding season. Tadanainen et al. found that rats (*Rattus norvegicus*) showed testicular regression with apoptotic characteristics after a lack of stimulation of gonadotropin secreted by the pituitary gland [63], providing a scientific reference for the biological mechanism of testicular regression in seasonally breeding animals. That team used the Djungarian hamster, also known as the Siberian hamster, as the research object and found that testicular regression during seasonal reproduction in animals was accompanied by an increase in cell apoptosis [64]. Subsequently, in many species, such as the Syrian hamster [65], white-footed mice (*Perimyscus leucopus*) [66], European hare (*Lepus europaeus*) [67], and domestic cat (*Felis catus*) [68], it was confirmed that cell apoptosis was the cause of testicular regression. However, Blottner et al. found that the seasonal testicular regression in roe deer (*Capreolus capreolus*) may be caused by the alternating activation and inhibition of GC proliferation and proposed for the first time that apoptosis is not the cause of testicular regression [69]. In addition, Dadhich et al. studied the Iberian mole (*Talpa occidentalis*) and found that, due to the low testosterone level in the testis, the SCs lost the care and support function for the GCs [70]. This was mainly manifested in the loss of the physiological function of the blood–testis barrier, resulting in the shedding of live, non-apoptotic GCs in the seminiferous tubules, which was considered a new mechanism of testicular regression during seasonal reproduction in animals [70]. Recently, the coordination of cell apoptosis and autophagy as well as the dedifferentiation of GCs have been found to cause testicular regression of seasonal reproduction in animals [10,71] (Table 1). At present, there is no scientific or systematic conclusion regarding the biological mechanism of testicular regression for seasonal reproduction in animals. Interestingly, AMH could promote the expression of apoptotic genes in testicular SCs and increase the apoptosis rate of cells [46]. Therefore, its physiological role in testicular regression during seasonal reproduction in animals cannot be ignored.

## 6. Potential Relationship between AMH and Seasonal Reproduction

The gonadal activity of seasonal reproduction undergoes periodic changes throughout the year. During the non-breeding season, gonadal regression reduces the energy consumption required for breeding activities to cope with adverse environmental conditions. In males, the weight of the testes is significantly lower during the non-breeding season, and spermatogenesis has a seasonal phenomenon in some animals. For example, in Syrian hamsters, spermatogenesis stagnates at the spermatocyte stage [62]. Spermatogenesis in the plateau pika (*Ochotona curzoniae*) and plateau zokor (*Eospalax baileyi*) is blocked at the undifferentiated spermatogonia stage [81,82]. During the non-breeding season, the convoluted tubules of the Japanese field mouse (*Apodemus speciosus*) only contained spermatogonia and a large number of apoptotic spermatocytes [83]. During the breeding season, testicular activity recovers, spermatogenesis begins, and mature sperm are gradually produced for breeding activities.

Owing to the low activity of the HPG axis before puberty in humans, the testicular tissue is mainly composed of SCs. The SCs are constantly proliferating under the regulation of FSH, resulting in a moderate increase in testicular volume; at the same time, the secretion level of AMH is high. Clinically, a testicular volume of 4 mL is defined as the beginning of puberty in the human testes [39]. During this period, due to the increase in HPG axis activity, the testosterone secretion level of the testicular tissue increases under the regulation of gonadotropin, and the hormone level of AMH gradually decreases, accompanied by the differentiation of GCs and a sharp increase in testicular volume. The changes in testicular tissue from pre-puberty to puberty are similar to those in seasonally breeding animals from the non-breeding season to the breeding season. Recently, a study pointed out that the testicular tissue of the plateau pika gradually regressed from June to July, the number of spermatogonia in each seminiferous tubule increased, most of the SC nuclei moved and gathered in the center of the seminiferous tubule, and the AMH protein began to be expressed in spermatogonia and SCs [10]. Similarly, our team performed transcriptomic assays on the testes of plateau zokors and found that the *AMH* gene had a high expression level during the non-breeding season and a low expression level during the breeding season [84]. These studies indicate that AMH may play a role in the testicular cycle during seasonal reproduction in animals.

The change in photoperiod is considered the main environmental factor that regulates seasonal reproduction. The MLT secreted by the pineal gland is strictly affected by light conditions. The activity of GnRH neurons in the hypothalamus is indirectly regulated by MLT, thereby affecting the production of LH and FSH in the pituitary gland. The serum LH and FSH levels of the plateau pika and plateau zokor were measured during the breeding and non-breeding seasons. It was found that there was no significant difference in the hormone secretion level of FSH, but the hormone level of LH during the breeding season was higher than that during the non-breeding season [85,86], indicating that the physiological effect of FSH on testicular SCs continued. The LH promoted the production of androgen (mainly testosterone) by testicular LCs during the breeding season, causing androgen to enter SCs and bind to the AR to inhibit the expression of the *AMH* gene; the inhibition effect was greater than the stimulation effect of FSH (Figure 4). The *AMH* gene is significantly inhibited during the breeding season, and this inhibition is accompanied by spermatogenesis during seasonal reproduction in animals. The hormone levels of LH and androgen decrease during the non-breeding season so that this inhibition is eliminated and the regulated expression of the *AMH* gene, which is dependent on FSH, increases. Therefore, AMH may play an important role in restricting testicular development and spermatogenesis during seasonal reproduction in animals, which has not been explored extensively in previous studies.

## 7. Conclusions and Future Direction

Animals have evolved different reproductive strategies to better adapt to complex living environments. Light, temperature, and precipitation are the external environmental factors that influence seasonal reproduction. Animals perceive these signals through receptors and transmit them to the brain to regulate HPG axis activity. Many hormones, such as GnRH, gonadotropin, and MLT, have been proven to play a crucial role in seasonal reproduction regulation. In mammals, AMH is involved in embryonic sex differentiation and reproductive regulation. However, it is unclear whether AMH plays a regulatory role in inhibiting testicular development and restricting spermatogenesis in seasonally breeding animals (Figure 5). In the future, studies are needed that, through hormone injection or over-expression of the gene, increase the level of serum AMH in mature animals during the breeding season and test whether it restricts the growth and development of testicular tissue and spermatogenesis. This research is of great significance to broadening knowledge on AMH’s physiological function in reproductive suppression and improving the theoretical basis of seasonal reproduction. Therefore, understanding the physiological functions and mechanisms of AMH in seasonal reproductive regulation can enrich the theory of seasonal reproductive regulation and guide the management of animal reproductive regulations. Moreover, it can be applied to the management and utilization of animal resources and provide a reference for the treatment of reproductive disorders caused by AMH.

## Figures and Tables

**Figure 1 ijms-24-05874-f001:**
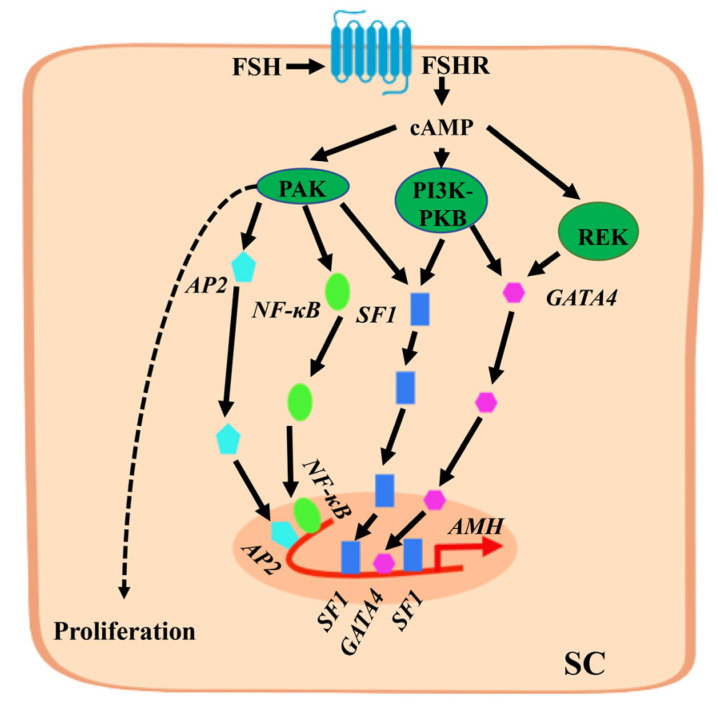
Mechanism by which follicle-stimulating hormone (FSH) stimulates anti-Müllerian hormone (*AMH*) gene expression in Sertoli cells (SCs). FSH binds to FSH receptor (FSHR) on SCs to induce an increase in cyclic adenosine monophosphate (cAMP) levels in the cytoplasm and activate protein kinase A (PKA), PI3K-PKB, REK, and other protein kinases. PKA is the main kinase involved in the up-regulation of *AMH* gene transcription, inducing the translocation of *SOX9*, *SF1*, and *GATA4* transcription factors to enter the nucleus and bind to the proximal promoter region. It also increases the level of *AP2* and *NF-κB* transcription factors in the nucleus and binds to the distal promoter region sequence of the *AMH* gene. They jointly promote the expression of the *AMH* gene. The proliferation of SCs is also promoted by cAMP through PKA.

**Figure 2 ijms-24-05874-f002:**
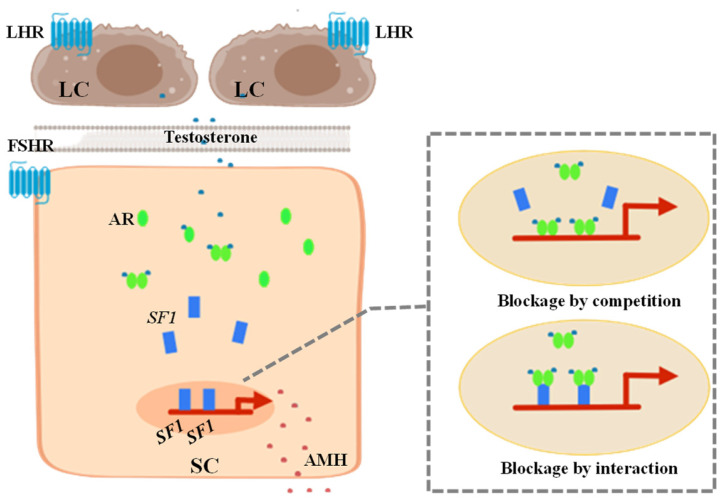
Mechanism of androgen receptor (AR) inhibiting the activity of anti-Müllerian hormone (*AMH*) gene promoter through *SF1* in Sertoli cells (SCs). Androgen produced by the Leydig cells (LCs) enters SCs through biofilm and binds to AR to form a dimer that inhibits *AMH* gene transcription. (1) Blockage by competition may be that AR and *SF1* compete for the binding site on the *AMH* gene promoter to prevent *SF1*-mediated gene up-regulation and achieve androgen inhibition of the activity of the *AMH* promoter. (2) Blockage by interaction may be that AR binding to a site on *SF1* changes the conformation of *SF1* and prevent it from exerting its stimulatory effect on *AMH* promoter activity enabling androgen-mediated downregulation.

**Figure 3 ijms-24-05874-f003:**
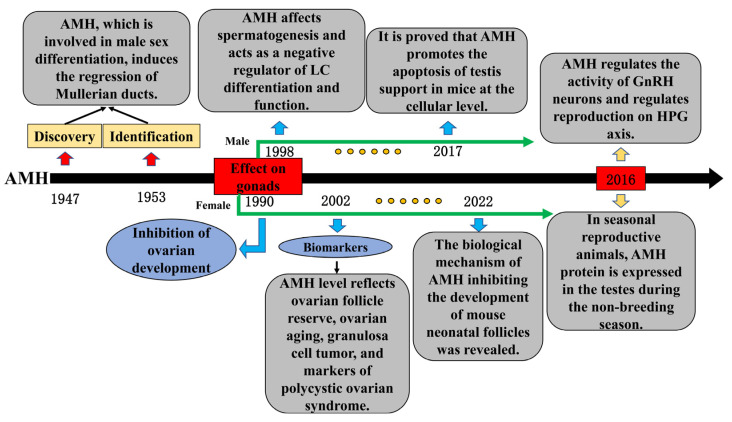
Important time nodes and research content in the progress of anti-Müllerian hormone (AMH) research.

**Figure 4 ijms-24-05874-f004:**
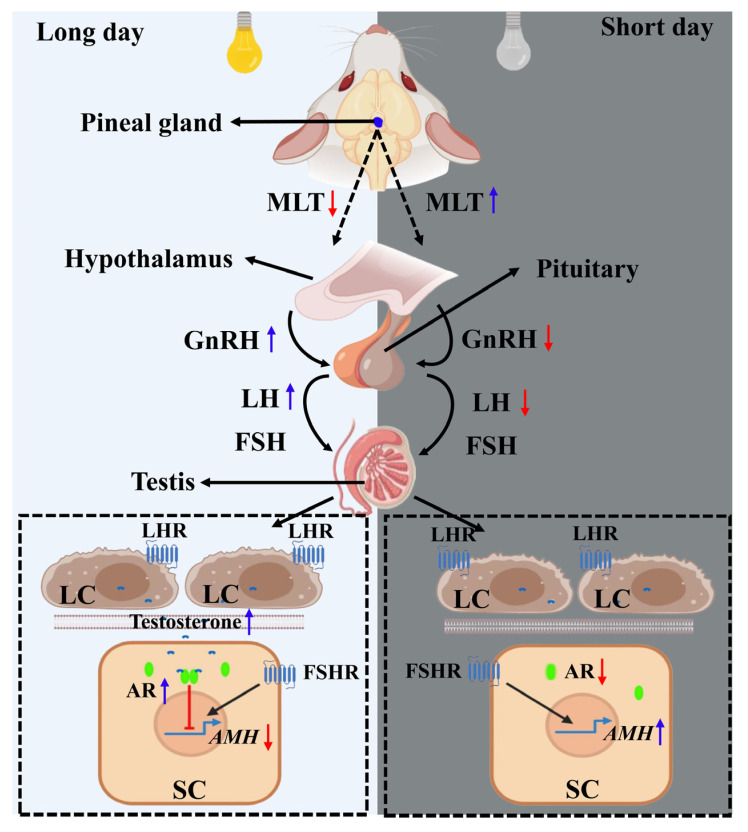
Potential relationship between anti-Müllerian hormone (AMH) and the seasonal reproduction pathway. For long-day breeders, the eyes sense the long day and send signals to reduce the secretion of MLT in the pineal gland during the breeding season. The low-level MLT indirectly promotes the activity of gonadotropin-releasing hormone (GnRH) neurons in the hypothalamus. The secreted GnRH stimulates the synthesis and secretion of luteinizing hormone (LH) and follicle-stimulating hormone (FSH) in the pituitary. LH acts on the Leydig cells (LCs) to promote the production of androgen, and the androgen enters the Sertoli cells (SCs) to inhibit the expression of the *AMH* gene by binding to the androgen receptor (AR), with the inhibition effect being greater than the promotion effect of FSH. In contrast, when the long-day breeder is in the non-breeding season (short day), the secretion of MLT in the pineal gland increases, which indirectly causes a decrease in the reduction in the activity of GnRH neurons in the hypothalamus and the synthesis and secretion of LH in the pituitary gland. This decreases the synthesis of androgen regulated by LH in the LCs, resulting in the weakening or disappearance of the inhibition of androgen on the expression of the *AMH* gene. However, the promotion of FSH continues and the *AMH* gene expression increases. The blue arrow indicates an increase in hormone secretion, while the red arrow indicates a decrease in hormone levels.

**Figure 5 ijms-24-05874-f005:**
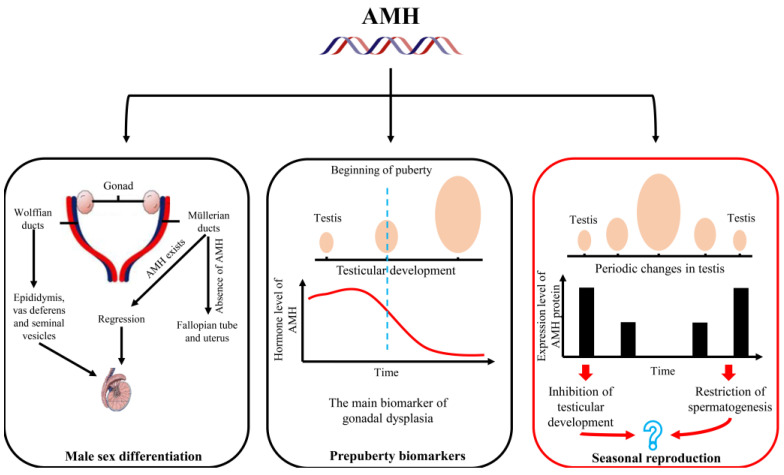
The main research fields of anti-Müllerian hormone (AMH) and its potential research value in the regulation of seasonal reproduction are summarized.

**Table 1 ijms-24-05874-t001:** Main research progress on the mechanism of testicular regression in mammals.

Species (*Latin name*)	Mechanism of Testicular Regression	Time	Reference
Rat (*Rattus norvegicus*)	Apoptosis	1993	[63]
		1995	[72]
		1996	[73]
Mouse (*Mus musculus*)		1997	[74]
Djungarian hamster (*Phodopus sungorus*)		1994	[64]
White-footed mouse (*Peromyscus leucopus*)		1999	[66]
		2000	[75]
		2001	[76]
Syrian hamster (*Mesocricetus auratus*)		2002	[65]
Brown hare (*Lepus europaeus*)		2003	[67]
Cat (*Felis catus*)		2022	[68]
Roe deer (*Capreolus capreolus*)	Alternating activation and inhibition of spermatogenic cell proliferation	2007	[69]
Iberian mole (*Talpa occidentalis*)	Cell desquamation (falling off or detaching)	2010	[70]
		2013	[77]
Hairy armadillo (*Chaetophractus villosus*)		2014	[78]
Egyptian long-eared hedgehog (*Hemiechinus auritus*)		2018	[79]
Mediterranean pine vole (*Microtus duodecimcostatus*)		2022	[80]
Plateau pika (*Ochotona curzoniae*)	Natural dedifferentiation process and autophagy	2016	[10]
Plains vizcacha (*Lagostomus maximus*)	Apoptosis and autophagy	2018	[71]

Note: Seasonally breeding animals are the main species in the table.

## Data Availability

Not applicable.

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
