# Peer review of "Potential Role of Anti-Müllerian Hormone in Regulating Seasonal Reproduction in Animals: The Example of Males"

_ijms, 2023, doi:10.3390/ijms24065874_

Round 1

Reviewer 1 Report

The review's topic is interesting and is certainly concerns the scientific community that studies the endocrine control of reproduction. Authors should pay more attention to formatting the manuscript according to the journal's requirements (such as placing reference numbers in square brackets along the manuscript). They also have to add their email address in the affiliation.

Author Response

Response to Reviewer 1 Comments

Comments and Suggestions for Authors

The review's topic is interesting and is certainly concerns the scientific community that studies the endocrine control of reproduction. Authors should pay more attention to formatting the manuscript according to the journal's requirements (such as placing reference numbers in square brackets along the manuscript). They also have to add their email address in the affiliation.

Response: Thank you for your pertinent suggestion. We have added reference numbers and email address to the manuscript.

Reviewer 2 Report

Authors reviewed the current research on the regulation of anti-mullerian hormone gene expression in seasonal male species and the regulatory pathways of seasonal reproduction with an emphasis on the association with AMH.  The information is very interesting but the English and grammar require extensive editing throughout the entire text. The figures are informative and helpful with visualization of gene regulation. There were other concerns about the presentation of the text listed below:

1.     Why are there numbers after words throughout the manuscript?  Was that supposed to be a list of something? Were these references?  If references then a parentheses needs to be around each number.

                                               i.      Ex. Line 33: ‘throughout the year 1……’

                                             ii.     Ex. Line 35: ‘success rate of breeding 2…’

                                            iii.     Ex. Line 38: ‘food resources 3…..’

                                            iv.     Examples throughout the rest of the text. Numbers everywhere with no designation.

2.     Line 51-57 and 211-220 and 222-227 and 239-232 and 282-286 and 289-296 and 348-353:  one long sentence that ends up lacking clarity, makes no sense.

3.     Line 59: AMH has already been defined earlier in the text. Do not redefine and do not begin a sentence with an acronym.

4.     Lines 132, 212, 222, 232, 296, 305, 308,: Valeri et al.; Meisohn et al.; Rehman et al.; Cimino et al. etc,  needs a reference number assigned or if references are alphabetical then the year in parentheses is required.

5.     Line 138: Acronym for estrogen receptor alpha is different than referenced in line 134.

Author Response

Response to Reviewer 2 Comments

Authors reviewed the current research on the regulation of anti-mullerian hormone gene expression in seasonal male species and the regulatory pathways of seasonal reproduction with an emphasis on the association with AMH.  The information is very interesting but the English and grammar require extensive editing throughout the entire text. The figures are informative and helpful with visualization of gene regulation. There were other concerns about the presentation of the text listed below:

  1. Why are there numbers after words throughout the manuscript? Was that supposed to be a list of something? Were these references? If references then a parenthesis needs to be around each number.
  2. Ex. Line 33: ‘throughout the year 1……’
  3. Ex. Line 35: ‘success rate of breeding 2…’

iii.  Ex. Line 38: ‘food resources 3…..’

  1. Examples throughout the rest of the text. Numbers everywhere with no designation.

Response: Thank you for your pertinent comment. The numbers in the manuscript represent the reference serial number, and we have made normative modifications according to the requirements of the journal.

  1. Line 51-57 and 211-220 and 222-227 and 239-232 and 282-286 and 289-296 and 348-353: one long sentence that ends up lacking clarity, makes no sense.

Line 51-57:

Response: Thank you for your valuable comment. We have made the necessary changes (Lines 59-62).

Line 211-220:

Response: Thank you for your suggestion. This sentence has been changed (Lines 208-211).

Line 222-227:

Response: Thank you very much for pointing this out. We have made the necessary modifications (Lines 213-216).

Line 229-232:

Response: Thank you for your suggestion. The descriptions here have been changed (Lines 222-225).

Line 282-286:

Response: Thank you for pointing this out. We have made changes (Lines 271-273).

Line 289-296:

Response: Thank you for your suggestion. We have made modifications (Lines 273-279).

Line 348-353:

Response: Thank you for your comment. The descriptions here have been changed (Lines 332-336).

  1. Line 59: AMH has already been defined earlier in the text. Do not redefine and do not begin a sentence with an acronym.

Response: Thank you very much for pointing this out. We have made the necessary changes and moved the sentences to lines 49-54 while ensuring that the manuscript logic is correct.

  1. Lines 132, 212, 222, 232, 296, 305, 308, Valeri et al.; Meisohn et al.; Rehman et al.; Cimino et al. etc, needs a reference number assigned or if references are alphabetical then the year in parentheses is required.

Response: Thank you for your pertinent suggestion. We have added reference numbers to the manuscript.

  1. Line 138: Acronym for estrogen receptor alpha is different than referenced in line 134.

Response: Thank you for your suggestion. We have made the necessary modification (Lines131).

Reviewer 3 Report

In this review entitled: “Potential Role of Anti-Mullerian Hormone in Regulating Seasonal Reproduction in Animals. On the Example of Males”, the authors summarized the research progress on the AMH gene expression, regulatory factors of gene expression and its role in reproductive regulation. They focused on males and in particular on testicular regression as a function of the seasonal regulatory path reproduction by assessing the potential relationship between AMH and seasonality reproduction, to broaden the physiological function of AMH in reproductive suppression, thereby providing new insights into understanding the regulatory pathway of seasonal reproduction. they conclude that understanding physiological functions and mechanisms of AMH in seasonal reproductive regulation may be useful in the treatment of reproductive disorders caused by AMH. This review is very interesting and can therefore provide useful information. In my opinion it can be accepted with minor revisions:

Line 73: the authors report that: “in XY embryos when the testicle and gonadal ridge differentiate”. The testis is known to arise from the gonadal crest, but as reported by the sentence this is not clear. Authors are invited to rephrase the sentence.

Lines: 87-89: this sentence is not clear. Authors are invited to rephrase the sentence.

Lines: 101-106. The sentence is too long and therefore unclear. Authors are invited to rephrase the sentence

Line 146: StAR. Authors are invited to indicate the meaning of the abbreviations the first time after their use and then use the abbreviations.

Line: 152: AR Authors are invited to indicate the meaning of the abbreviations the first time after their use and then use the abbreviations.

Line 167: NF-κB Authors are invited to indicate the meaning of the abbreviations the first time after their use and then use the abbreviations.

Line 182: SF1 Authors are invited to indicate the meaning of the abbreviations the first time after their use and then use the abbreviations.

Line 208: AIS Authors are invited to indicate the meaning of the abbreviations the first time after their use and then use the abbreviations.

Line 211: “Previous studies….” since it is a review, the bibliographic reference is very important. Authors may add references.

Lines 212-220. The sentence is too long and therefore unclear. Authors are invited to rephrase the sentence.

Line 220: In male species, Early studies….correct with early……

Lines 222-227: The sentence is too long and therefore unclear. Authors are invited to rephrase the sentence.

Lines 227-228: the authors report:  In addition, studies have shown that AMH can play a regulatory role in the HPG axis during reproduction. Can the authors report in which species.

Line 231: PCOS.  Authors are invited to indicate the meaning of the abbreviations the first time after their use and then use the abbreviations.

Line 255: TH exists in four forms in animals… Can the authors report in which animals

Author Response

Response to Reviewer 3 Comments

Comments and Suggestions for Authors

In this review entitled: “Potential Role of Anti-Mullerian Hormone in Regulating Seasonal Reproduction in Animals. On the Example of Males”, the authors summarized the research progress on the AMH gene expression, regulatory factors of gene expression and its role in reproductive regulation. They focused on males and in particular on testicular regression as a function of the seasonal regulatory path reproduction by assessing the potential relationship between AMH and seasonality reproduction, to broaden the physiological function of AMH in reproductive suppression, thereby providing new insights into understanding the regulatory pathway of seasonal reproduction. they conclude that understanding physiological functions and mechanisms of AMH in seasonal reproductive regulation may be useful in the treatment of reproductive disorders caused by AMH. This review is very interesting and can therefore provide useful information. In my opinion it can be accepted with minor revisions:

  1. Line 73: the authors report that: “in XY embryos when the testicle and gonadal ridge differentiate”. The testis is known to arise from the gonadal crest, but as reported by the sentence this is not clear. Authors are invited to rephrase the sentence.

Response: Thank you very much for pointing out this important issue. We have made the necessary modifications (Lines 72-74).

  1. Lines: 87-89: this sentence is not clear. Authors are invited to rephrase the sentence.

Response: We apologize for the unclear description in our manuscript. We have made the necessary changes (Lines 88-89).

  1. Lines: 101-106. The sentence is too long and therefore unclear. Authors are invited to rephrase the sentence

Response: Thank you for your pertinent suggestion. We have given a brief description of the sentence (Lines 100-103).

  1. Line 146: StAR. Authors are invited to indicate the meaning of the abbreviations the first time after their use and then use the abbreviations.

Response: Thank you for your comment. The relevant instructions have been added to the revised article (Line 142).

  1. Line: 152: AR Authors are invited to indicate the meaning of the abbreviations the first time after their use and then use the abbreviations.

Response: Thank you very much for pointing this out. The meaning of AR has been indicated in line 122 of the manuscript.

  1. Line 167: NF-κB Authors are invited to indicate the meaning of the abbreviations the first time after their use and then use the abbreviations.

Response: Thank you for your pertinent suggestion. The meaning of NF-κB has been indicated in line 101 of the manuscript.

  1. Line 182: SF1 Authors are invited to indicate the meaning of the abbreviations the first time after their use and then use the abbreviations.

Response: Thank you for your comment. The meaning of SF1 has been indicated in line 81 of the manuscript.

  1. Line 208: AIS Authors are invited to indicate the meaning of the abbreviations the first time after their use and then use the abbreviations.

Response: Thank you for your suggestion. The meaning of AIS has been indicated in line 120 of the manuscript.

  1. Line 211: “Previous studies….” since it is a review, the bibliographic reference is very important. Authors may add references.

Response: Thank you very much for pointing this out. We have added a relevant reference (Lines 208).

  1. Lines 212-220. The sentence is too long and therefore unclear. Authors are invited to rephrase the sentence.

Response: Thank you for your pertinent suggestion. We have made the necessary changes (Lines 208-211).

  1. Line 220: In male species, Early studies….correct with early……

Response: Thank you very much for pointing this out. We have made the necessary modification (Lines 212).

  1. Lines 222-227: The sentence is too long and therefore unclear. Authors are invited to rephrase the sentence.

Response: Thank you for your valuable comment. We have made the necessary changes (Lines 213-216).

  1. Lines 227-228: the authors report: In addition, studies have shown that AMH can play a regulatory role in the HPG axis during reproduction. Can the authors report in which species.

Response: We apologize for the inaccuracy description in our manuscript. We have made the necessary modification (Lines 217-218).

  1. Line 231: PCOS. Authors are invited to indicate the meaning of the abbreviations the first time after their use and then use the abbreviations.

Response: Thank you very much for pointing this out. The meaning of PCOS has been indicated in line 218 of the manuscript.

  1. Line 255: TH exists in four forms in animals… Can the authors report in which animals

Response: Thank you for your valuable comment. We have made the necessary changes for the sentence to ensure its accuracy, and we added the reported animals (Lines 244-246).

Round 2

Reviewer 2 Report

Authors have made significant revisions to the english grammar, which is appreciated.  Only a few minor corrections are required.

1.     Line 151: Replace the word ‘puberty’ with ‘pubertal’

2.     Line 187-189: Do not end the sentence after ref 37. End it after ‘immune cells’. Begin the next sentence with ‘However’.

3.     Line 227: delete ‘s’ from the word ‘affect’

4.     Line 243: add an ‘s’ to the word ‘males’

5.     Line 350:replace the word ‘testicles’ with the word ‘testicular’

6.     Line 352: delete the ‘s’ in the word ‘increases’

7.     Figure 4: cell pictures require improved resolution; each pic appears out of focus, fuzzy.

8.     Line 435-442: one long sentence and it doesn’t make any sense. Revise sentence.

Author Response

Comments and Suggestions for Authors

Authors have made significant revisions to the English grammar, which is appreciated.  Only a few minor corrections are required.

  1. Line 151: Replace the word ‘puberty’ with ‘pubertal’

Response: Thank you very much for pointing out this important issue. We have made the necessary modifications (Lines 130-131).

  1. Line 187-189: Do not end the sentence after ref 37. End it after ‘immune cells’. Begin the next sentence with ‘However’.

Response: Thank you for your pertinent suggestion. We have changed (Lines 164-166).

  1. Line 227: delete ‘s’ from the word ‘affect’

Response: Thank you for your valuable comment. We have modified (Lines 204).

  1. Line 243: add an ‘s’ to the word ‘males’

Response: Thank you for pointing out this important issue. We have made the necessary changes (Lines 212).

  1. Line 350: replace the word ‘testicles’ with the word ‘testicular’

Response: Thank you for your valuable comment. We have made the necessary modifications (Lines 301).

  1. Line 352: delete the ‘s’ in the word ‘increases’

Response: Thank you very much for pointing out this important issue. We have modified (Lines 303).

  1. Figure 4: cell pictures require improved resolution; each pic appears out of focus, fuzzy.

Response: Thank you for your pertinent suggestion. We have improved resolution of the picture (Lines 360).

  1. Line 435-442: one long sentence and it doesn’t make any sense. Revise sentence.

Response: Thank you for your valuable comment. We have given a brief description of the sentence (Lines 385-390).
